# A Novel CreA-Mediated Regulation Mechanism of Cellulase Expression in the Thermophilic Fungus *Humicola insolens*

**DOI:** 10.3390/ijms20153693

**Published:** 2019-07-28

**Authors:** Xinxin Xu, Chao Fan, Liya Song, Jinyang Li, Yuan Chen, Yuhong Zhang, Bo Liu, Wei Zhang

**Affiliations:** 1Biotechnology Research Institute, Chinese Academy of Agricultural Sciences, No.12 Zhongguancun South St., Haidian District, Beijing 100081, China; 2Beijing Key Lab of Plant Resource Research and Development, Beijing Technology and Business University, No.11 Fucheng Road, Haidian District, Beijing 100048, China

**Keywords:** cellulase, *creA*, *Humicola insolens*, thermophilic fungi

## Abstract

The thermophilic fungus *Humicola insolens* produces cellulolytic enzymes that are of great scientific and commercial interest; however, few reports have focused on its cellulase expression regulation mechanism. In this study, we constructed a *creA* gene (carbon catabolite repressor gene) disruption mutant strain of *H. insolens* that exhibited a reduced radial growth rate and stouter hyphae compared to the wild-type (WT) strain. The *creA* disruption mutant also expressed elevated pNPCase (cellobiohydrolase activities), pNPGase (β-glucosidase activities), and xylanase levels in non-inducing fermentation with glucose. Unlike other fungi, the *H. insolens*
*creA* disruption mutant displayed lower FPase (filter paper activity), CMCase (carboxymethyl cellulose activity), pNPCase, and pNPGase activity than observed in the WT strain when fermentation was induced using Avicel, whereas its xylanase activity was higher than that of the parental strain. These results indicate that CreA acts as a crucial regulator of hyphal growth and is part of a unique cellulase expression regulation mechanism in *H. insolens*. These findings provide a new perspective to improve the understanding of carbon catabolite repression regulation mechanisms in cellulase expression, and enrich the knowledge of metabolism diversity and molecular regulation of carbon metabolism in thermophilic fungi.

## 1. Introduction

The thermophilic fungus *Humicola insolens* produces an array of enzymes to degrade cellulose and hemicellulose [1,2,3,4,5]. Compared to mesophilic fungi, *H. insolens* exhibits a higher optimal growth temperature (45 °C), faster growth rate, and stronger secretion capacity of cellulases [6]. Cellulases produced by *H. insolens* are active under high temperatures and neutral conditions, and exhibit good thermostability and a wide pH adaptation range, resulting in great application potential in biomass degradation and in the textile, brewing, and feed industries [2,6,7,8]. *H. insolens* also can produce large amounts of foreign proteins. Thus, *H. insolens* is expected to become an excellent cellulase industrial production strain and protein expression system. However, its genetic background remains unclear and few studies have explored its cellulase expression regulation mechanisms [5,9].

Cellulase gene transcription in filamentous fungi has been shown to be regulated by several transcription factors, including CreA (carbon catabolite repressor A), Ace1 (activator of cellulases 1), Xyr1 (xylanase regulator 1), Ace2 (activator of cellulases 2), and the Hap (heme activator protein) 2/3/5 complex [10,11,12,13,14]. CreA, a core responder in carbon metabolite repression (CCR), has been studied widely as it is a major transcription factor involved in cellulase and hemicellulase expression regulation in various filamentous fungi, including *Trichoderma reesei* [15], *Neurospora crassa* [16], *Aspergillus nidulans* [17], and *Penicillium oxalicum* [18]. In filamentous fungi, CreA inhibits cellulase expression at the transcriptional level when glucose is present in the medium. Apart from directly regulating cellulase gene transcription, CreA also affects transcriptional activators, such as Xyr1, and cellulase transporters [19,20]. Deletion of the *creA* gene in filamentous fungi has been shown to significantly improve cellulase production [18,21,22].

In this study, to explore the regulation by *creA* gene during *H. insolens* cellulase production, we constructed a *creA* gene disruption mutant strain and a *creA* gene complement strain. This is the first report of gene knockout in *H. insolens*. We also explored *creA* gene function in *H. insolens* through an intensive analysis of differences in phenotypes, hyphal morphology, and cellulase production between the wild-type (WT) strain, the *creA* gene disruption mutant, and *creA* complement strains. Findings from this study enrich our understanding of the molecular mechanisms of efficient lignocellulose decomposition by *H. insolens*, and constitute a valuable foundation for its further genetic improvement to enhance cellulase production.

## 2. Results

### 2.1. Isolation and Sequence Analysis of creA from H. insolens Y1

A 1260-bp, full-length *creA* gene was isolated from *H. insolens* Y1; it encoded a predicted polypeptide of 419 amino acids in length. The deduced CreA protein contains two C_2_H_2_ Zn-finger domains (residues 66–86 and 94–116). Based on sequence alignment analysis, the deduced CreA amino acid sequence shares 100, 56, 50, and 44% identity with CreA proteins from *Humicola grisea* (O93781), *N. crassa* (O59958), *T. reesei* (G0RB08), and *A. nidulans* (Q01981), respectively (Figure 1).

### 2.2. Disruption and Complementation of creA in H. insolens Y1

To determine the gene function of *creA* in *H. insolens* Y1, *creA* was knocked out using positive and negative screening methods with geneticin and hygromycin B resistance genes as selection markers (Figure 2a). Transformants that had acquired resistance to geneticin but could not grow in the presence of hygromycin B were selected as potential *creA* disruption mutants. Genomic DNA was extracted from the potential mutants, and primers within and outside the *creA* gene were used to verify that disruption had occurred. A 558-bp specific fragment was amplified by polymerase chain reaction (PCR) from the *H. insolens* WT strain, but not from the *creA* disruption mutant, using the *creA*IS/*creA*IA primer pair; 4742- and 3171-bp specific fragments were amplified from the *creA* disruption mutant and WT strains, respectively, using the *creA*S/*creA*A primer pair (Figure 2b). In addition, absolute quantitative PCR was performed to analyze the insertion copy number of the deletion cassette in the *creA* disruption mutant, and the results confirmed the single insertion of the deletion cassette (data not shown). To further determine the function of *creA* in *H. insolens*, *creA* complement strains were also constructed by introducing the entire *creA* gene into the disruption mutant.

### 2.3. Disruption of creA Affected the H. insolens Phenotype and Mycelium Morphology

To determine the effect of the *creA* gene on the *H. insolens* phenotype and mycelium morphology, the WT *H. insolens* Y1 strain, the *creA* disruption mutant, and the *creA* complement strains were cultured on modified Melin–Norkrans (MMN) and potato dextrose agar (PDA) plates. After three days of growth at 42 °C, no substantial differences in colony size or mycelium morphology were observed between the WT and *creA* complement strains, whereas the *creA* disruption mutant exhibited slower radial growth and smaller colonies than the WT (Figure 3a). The *creA* disruption mutant showed stouter hyphae and more branches on PDA plates at 24 h under microscopic observation (Figure 3b). 

### 2.4. Disruption of creA Revealed a Novel Regulation Mechanism for H. insolens Cellulase Production

To verify the role of the *creA* gene in *H. insolens* cellulase production, cellulase production by the WT, the *creA* disruption mutant, and the *creA* complement strains grown using shake-flask fermentation were evaluated. At 24 h in yeast extract peptone dextrose (YPD) medium, the FPase activity of the *creA* disruption mutant was similar to that of the WT and *creA* complement strains; CMCase activity was observed to follow a similar trend. In contrast, pNPCase, pNPGase, and xylanase activities were 518.35, 119.58, and 42.68% higher than that of the WT strain, respectively (Figure 4). 

The levels of cellulase activity and secreted proteins in MMN medium were recorded over a period of six days. Throughout the fermentation period, no significant difference was detected between the parental and *creA* complement strains (Figure 5). In contrast, the biomass of the *creA* disruption mutant was slightly lower than that of the WT strain at five to six days of fermentation (Figure 5a). The *creA* disruption mutant exhibited lower FPase activity than the WT strain after four days of cultivation. After six days of fermentation, the maximum FPase activity of the *creA* disruption mutant was observed at 7.89 U/mL, representing 80.43% of the 9.81 U/mL shown by the WT strain (Figure 5b). The extracellular CMCase, pNPCase, pNPGase, and xylanase activities were also determined. The maximum CMCase, pNPCase, and pNPGase activities in the *creA* disruption mutant were significantly reduced to 70.40, 71.22, and 41.58% of that of the parental strain, respectively (Figure 5c–e). In contrast, the xylanase activity was 21.97% higher than that of the parental strain (Figure 5f).

Extracellular protein secretion profiles were then examined by SDS-PAGE. The protein yield of the *creA* disruption mutant was significantly greater than that of the WT strain in YPD medium fermentation (Appendix A). SDS-PAGE results showed a significant decrease in the total secreted proteins of the *creA* disruption mutant; notably, the *creA* disruption mutant band at around 40 kDa was much weaker than that of the WT strain in MMN medium fermentation (Appendix A).

### 2.5. Regulation of Cellulolytic Gene Expression by CreA in H. insolens

qRT-PCRs were performed to determine the influence of *creA* disruption on the transcriptional level of cellulolytic genes. When cultured for 24 h in YPD medium, the transcriptional levels of all the target genes (including the β-glucosidase genes *bgl3B and bgl3C,* cellobiohydrolase genes *cel6A* and *cel7A*, endoglucanase genes *cel6B* and *cel7B*, and xylanase genes *xynA* and *xynC*) were enhanced in the *creA* disruption mutant except *bgl3A* and *xynB* (Figure 6a). 

The transcriptional levels of cellulolytic genes in the *creA* disruption mutant and WT *H.insolens* strain under inducing condition were also investigated. Consistent with the enzyme activities, all the xylanase genes—*xynA, xynB*, and *xynC*—showed much higher mRNA levels in the *creA* disruption mutant than that in WT strain when the strains were cultured for 48 h in MMN medium (Figure 6b). On the contrary, the transcriptional levels of all the cellulase genes except *cel7A* were significantly decreased in the *creA* disruption mutant. 

## 3. Discussion

*H. insolens* is a potential industrial cellulase producer due to its powerful ability to degrade polysaccharide biomass constituents such as cellulose and hemicellulose. To date, research on *H. insolens* has mainly focused on the purification of extracellular enzymes and the isolation and heterologous expression of related carbohydrate hydrolases, with few reports on the molecular mechanisms of cellulase expression regulation. In the present study, we performed the first reported knockout of the *creA* gene and determined its function in *H. insolens*. The results of this study provide further opportunities for studying the functions of *H. insolens* genes, and will help to reveal their cellulase regulation mechanisms.

CreA from *H. insolens* shows high identity with those from *H. grisea*, *N. crassa*, *T. reesei*, and *A. nidulans*. As previously shown in *creA* disruption strains of *Penicillium oxalicum*, *Penicillium expansum,* and *T. reesei* [18,22,23,24], the *H. insolens creA* disruption strain significantly affected hyphal growth rate and development. Previous findings have suggested that CreA affects morphology and sporulation by interacting with RAS1 in *T. reesei* [23]. As a regulator, RAS1 affects hyphal morphology, reproductive development, and cell wall integrity in *N. crassa* and *A. fumigatus* through interaction with related genes [25,26]. The results of transcriptome analysis in *P. expansum* suggest that the alteration of carbon/nitrogen metabolism in the Δ*creA* mutant may contribute to the deviant fungal growth [24].

CreA also plays an important role in cellulase and hemicellulase production in *H. insolens*. In YPD medium fermentation with glucose as a carbon source, the extracellular protein yield and pNPCase, pNPGase, and xylanase activities were higher in the *creA* disruption mutant than in the WT strain. These results suggest that *creA* disruption results in de-repression of cellulase and hemicellulase gene expression under non-inducing conditions. In contrast to results reported for other fungi [16,18,22], the *H. insolens creA* disruption mutant strain cultured under Avicel-inducing conditions exhibited significantly lower cellulase expression but higher xylanase production than the WT strain. The transcriptional levels of cellulolytic genes in *creA* disruption mutant and WT *H. insolens* strain under non-inducing and inducing conditions also matched well with the cellulase activities. These results indicate that *H. insolens* CreA plays a similar role in hemicellulase production to that in cellulase production, via a different regulation mechanism. CreA may inhibit cellulase gene transcription in the presence of glucose, and may also be indispensable for cellulase hyperproduction in *H. insolens*. We speculate that these results may be partially due to the influence of *creA* disruption on Xyr1, an important transcription cellulase and hemicellulose activator. Competition between CreA and Xyr1 for binding to sites on cellulase gene promoters has been reported [27,28,29]. In the *creA* disruption mutant, hemicellulase gene promoters recruited more Xyr1, in turn increasing their transcriptional levels. In contrast, the expression of *xyr1* decreased when *creA* was deleted in *H. insolens*, as previously described in *T. reesei* [20]. Limited Xyr1 preferentially activated hemicellulases such as xylanase; however, the difficulty of fully activating cellulases resulted in a reduction in cellulase production. In addition, due to the complexity of the regulation network of non-glucose source utilization, *creA* disruption may exert a complex influence on many regulation elements and transporters involved in cellulase expression. In future work, we plan to perform RNA sequencing, electrophoretic mobility shift assays (EMSAs), and DNaseI footprinting assays to elucidate the specific regulation mechanisms for *creA* in *H. insolens*.

## 4. Materials and Methods 

### 4.1. Strains, Media, and Culture Conditions

*H. insolens* strain Y1 CGMCC 4573 was cultured on potato dextrose agar (PDA) plates for 7 days at 42 °C to induce sporulation. Shake-flask fermentation was conducted at 42 °C for 6 days in modified Melin–Norkrans (MMN) medium (0.1% tryptone, 2% yeast extract, 0.06% MgSO_4_·7H_2_O, 2% Avicel). Yeast extract peptone dextrose (YPD) medium (2% tryptone, 1% yeast extract, 2% glucose) was used to culture hyphae. *Agrobacterium tumefaciens* strain AGL-1 grown at 28 °C was used to mediate DNA transformation of *H. insolens* Y1.

### 4.2. Molecular Manipulation

*H. insolens* Y1 genomic DNA was extracted from mycelia grown on PDA plates. The primers used in this experiment are listed in Appendix A. Polymerase chain reaction (PCR) amplification was performed using an Eppendorf Mastercycler and KOD FX Polymerase (Toyobo). Primer synthesis and DNA sequencing were performed by Sangon Biotech Corporation (Shanghai, China). 

### 4.3. Sequence Analysis

The predicted amino acid sequence of the CreA protein was analyzed using FGENESH (http://linux1.softberry.com/berry.phtml). An online conserved domain search tool (https://www.ncbi.nlm.nih.gov/Structure/cdd/wrpsb.cgi) and InterProScan (http://www.ebi.ac.uk/Tools/pfa/iprscan/) were used to analyze the CreA conserved domains and protein functions. Protein sequence alignment was performed using ClustalW (http://www.ebi.ac.uk/clustalW/).

### 4.4. Construction of H. insolens Y1 creA Disruption Mutant and Complementation Strains

The entire P*gpd*-*neo*-T*gpd* cassette was excised from pP-*neo*-T using *Sma*I and *Xho*I subjected to end filling, and then ligated into the *Swa*I site of pAg1-*hyg* [9] to yield pAg1-*HygNeo*. The *creA*L fragment was amplified from WT *H. insolens* genomic DNA using the *creA*LS/*creA*LA primer pair, and then ligated into the *Bgl*II site of pAg1-*Hyg*-*Neo* to generate pAg1-*HygNeo*-*creA*L. The *creA*R fragment was amplified in the same manner using the *creA*RS/*creA*RA primer pair, and then ligated into the *Pac*I site of pAg1-*HygNeo*-*creA*L to generate pAg1-*HygNeo*-Δ*creA*. pAg1-*HygNeo*-Δ*creA* was then transformed into an *H. insolens* strain using a *Agrobacterium tumefaciens*-mediated transformation (ATMT) system as previously described [9] to obtain *creA* disruption mutants.

The *creA* WT allele complementation cassette was amplified from WT *H. insolens* genomic DNA using the *creA*S/*creA*A primer pair. The *creA* cassette was then ligated into the *Pac*I site of pAg1-*hyg* to yield pAg1-*Hyg*-*creA*C. The resulting pAg1-*Hyg*-*creA*C was transformed into a *creA* disruption mutant to obtain *creA* complement strains. The sequences of all of the primers used in this study are shown in Appendix A.

### 4.5. Phenotypic Observation and Microscopic Observation

*H. insolens* WT, the *creA* disruption mutant, and *creA* complementation strains were cultured following inoculation of 2 × 10^5^ spores onto MMN, PDA, and YPD plates for 3 days at 42 °C, and the phenotypes of the resulting colonies were observed. Hyphal morphology was observed by microscopy (Eclipse Ni-U, Nikon, Tokyo, Japan) after spore germination or 24 h on cellophane-coated PDA and YPD plates.

### 4.6. Enzyme Activity Assay and Sodium Dodecyl Sulfate–Polyacrylamide Gel Electrophoresis (SDS–PAGE)

*H. insolens* WT, the *creA* disruption mutant, and *creA* complement strains were each cultured, starting with inoculation of the same number of spores, in 40 mL YPD medium for 24 h at 42 °C. Homogeneous medium (1 mL) was sampled from the YPD medium to detect enzyme activity and protein expression. After centrifuging at 12,000 rpm for 10 min and washing with ddH_2_O, the thalli in the remaining culture medium were transferred to MMN medium for shake-flask fermentation at 42° C for 6 days. The cellulase activities in the supernatant of each strain were assayed every 24 h. Filter paper cellulase (FPase), carboxymethylcellulose cellulose (CMCase), *p*-nitrophenyl-β-cellobioside cellulase (pNPCase), *p*-nitrophenyl-β-glucopyranoside cellulase (pNPGase), and xylanase enzyme activities were assayed as described [30]. 

SDS–PAGE was performed using 12% (*w/v*) polyacrylamide gels. Proteins were stained with Coomassie Brilliant Blue G-250. The *H. insolens* biomass in the MMN fermentation medium was measured as previously described [27]. 

### 4.7. Real-time Quantitative PCR Analysis

Absolute quantitative PCR was performed to analyze the integrated copy number of the deletion cassette in disruption mutant according to the reported method [31,32,33]. The copy number of the G418 resistance gene (*neo*) in *creA* disruption cassette in the *creA* disruption mutant was determined. The standard curve was measured using a plasmid (M13-PTgpd-neo) which contains single *neo* gene as the template.

The relative transcriptional level of cellulytic genes in *H. insolens* was determined by qRT-PCR. β-actin gene was used as an internal control for normalization. Reactions were performed using 2×ChamQ Universal SYBR qPCR Master Mix kit (Vazyme, Nanjing, China) as described by the manufacturer’s protocol. Data analysis was done with the 2^−△△Ct^ method.

The qPCR experiment was performed on the LightCycler 96 System as follows: 1 cycle of 94 °C for 5 min, followed by 40 cycles of 94 °C for 5 s, 58 °C for 15 s, and 72 °C for 20 s. The primers for qPCR are shown in Appendix A. Three biological replicates were performed with three technical replicates for each biological replicate.

### 4.8. Nucleotide Sequence

The nucleotide sequence of the *creA* gene of *H. insolens* Y1 was deposited into the GenBank database under accession number MK907677.

## Figures and Tables

**Figure 1 ijms-20-03693-f001:**
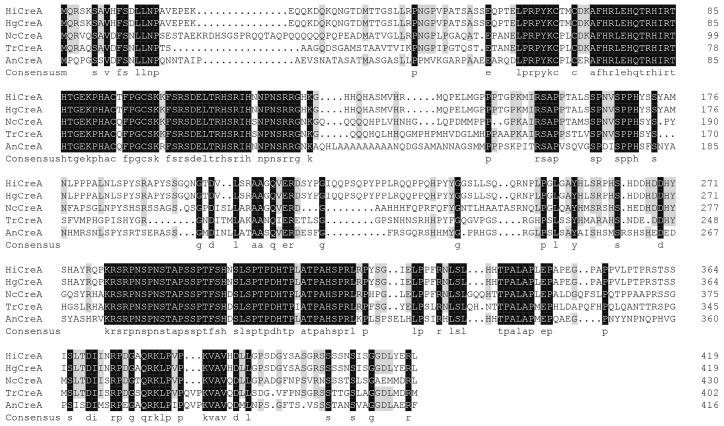
Multiple protein sequence alignment of the deduced CreA sequence from *H**. insolens* with four CreA counterparts. The sequences shown are CreA from *H. insolens*, HgCreA from *H. grisea* (O93781), NcCreA from *N. crassa* (O59958), TrCreA from *T. reesei* (G0RB08), and AnCreA from *A. nidulans* (Q01981). Identical and conserved residues are shaded in black and gray, respectively.

**Figure 2 ijms-20-03693-f002:**
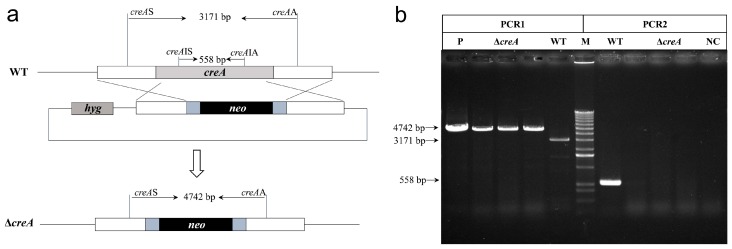
Verification of the *creA* disruption mutant. (**a**) Schematic diagram of the *creA* gene knockout mediated by homologous recombination. *hyg*, hygromycin B resistance gene expression cassette; *neo*, geneticin resistance gene expression cassette. (**b**) Polymerase chain reaction (PCR) analysis of the *creA* disruption mutant using the primer pairs *creA*S/*creA*A (PCR1) and *creA*IS/*creA*IA (PCR2). M, 1000-bp DNA ladder; WT, wild type; Δ*creA*, *creA* disruption mutant; P, positive control with the plasmid pAg1-*HygNeo*-Δ*creA* as the template; NC, negative control.

**Figure 3 ijms-20-03693-f003:**
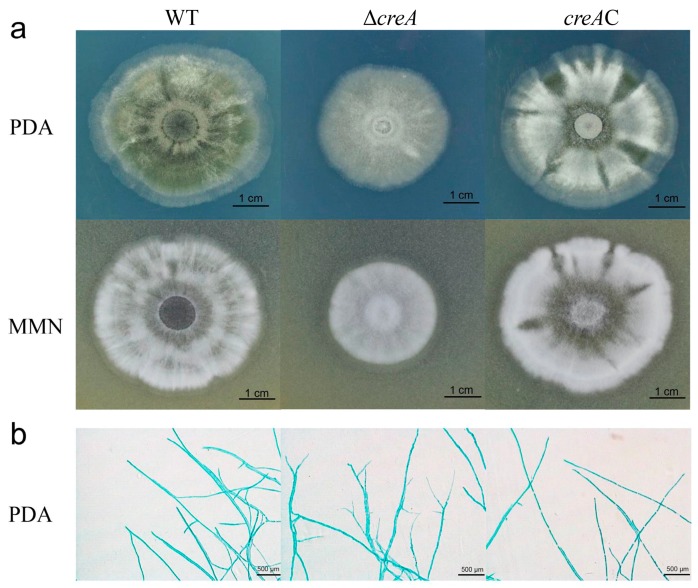
Characterization of the *creA* disruption mutant growth and hyphal morphology. (**a**) Colony growth and sporulation of the *creA* disruption mutant, *creA* complement strains, and the wild-type (WT) strain on modified Melin–Norkrans (MMN) and potato dextrose agar (PDA) plates at 42 °C for 3 days. (**b**) Hyphae observed under a microscope. All strains were grown on PDA plates at 42 °C for 24 h. *creA*C, *creA* complement strain.

**Figure 4 ijms-20-03693-f004:**
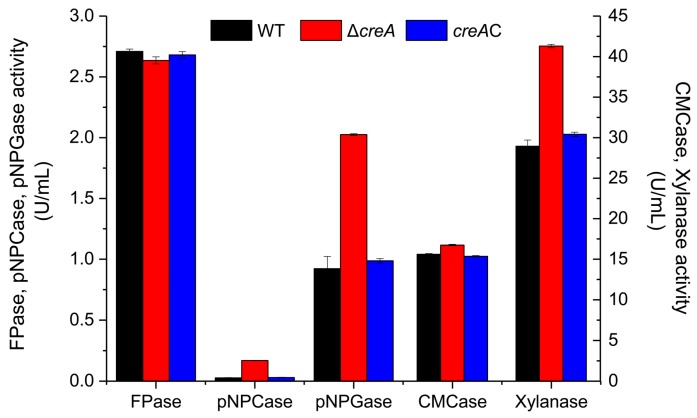
Cellulase activity of *H. insolens* WT, the *creA* disruption mutant, and *creA* complement strains in the presence of glucose. All strains were cultured in shake flasks at 42 °C for 24 h with 2.0% (*w*/*v*) glucose as the carbon source.

**Figure 5 ijms-20-03693-f005:**
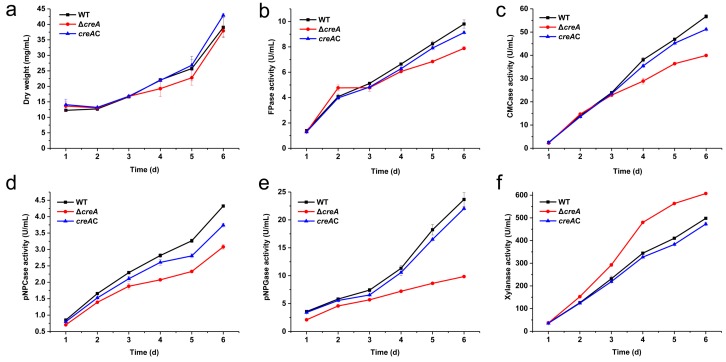
Cellulase activities of *H. insolens* WT, *creA* disruption mutant, and *creA* complement strains under inducing conditions. Dry weight (**a**), FPase activity (**b**), CMCase activity (**c**), pNPCase activity (**d**), pNPGase activity (**e**), and xylanase activity (**f**) were measured from shake-flask fermentation cultures grown at 42 °C for 6 days using 2.0% (*w/v*) Avicel as the carbon source.

**Figure 6 ijms-20-03693-f006:**
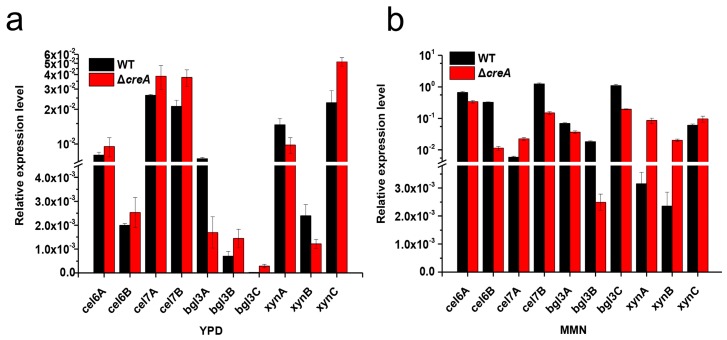
Relative transcriptional level of cellulolytic genes in *H. insolens* WT and *creA* disruption mutant under non-inducing (**a**) or inducing (**b**) conditions.

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
