# Peer review of "A Novel CreA-Mediated Regulation Mechanism of Cellulase Expression in the Thermophilic Fungus Humicola insolens"

_ijms, 2019, doi:10.3390/ijms20153693_

Round 1
Reviewer 1 Report
This manuscript reports on the characterization of the carbon catabolite repressor CreA in the fungus Humicola insolens. Findings from this study showed that CreA influences and regulates growth and other physiological traits in the fungus as well as cellulose expression mechanism. Overall, the manuscript contains interesting findings reported for the first that might be of great interest for International Journal of Molecular Science’ readers. Some minor English edits and analyses should be done before the acceptance of the manuscript for publication:
1. Line 19: Replace “with” by “to”
2. Line 35: A secretion of what?
3. Line 45: Do you mean “hemicellulase” not “hemicellulose”?
4. Line 52: Replace “of the creA” by “by creA”.
5. Line 57: Replace “complementary” by “complement” in the whole manuscript.
6. Line 57: Replace “The findings of this study” by “Findings from this study”.
7. Line 58: Replace “form” by “constitute”.
8. Line 86: Fix the typo “hgy”.
9. Why the authors have not done southern analysis to confirm the single insertion of the deletion cassette? I suggest to include these analyses.
10. In paragraph 2.4: Authors should define the enzymes’ abbreviations for their first appearance in the text.
11. Lines 110-111: Rephrase the sentence ‘’in contrast… (Figure 4)”.
12. Line 124: An “of” is missed after “41.58%”.
13. Line 147: Replace “of the molecular” by “on the molecular”.
14. Lines 152-158: A recent paper by Tannous et al. 2018, described the effect of CreA on the morphology and physiological traits in P. expansum. I suggest that the authors should include these findings and others in their discussion.
Reviewer 2 Report
The manuscript “A Novel CreA-mediated Regulation Mechanism of Cellulase Expression in the Thermophilic Fungus Humicola insolens” describes the consequences of knock-out of a cellulose/hemicellulose transcription factor for cellulose and xylanase expression and hyphal growth rate and morphology under non-inducing and inducing conditions. The study appears to be generally well-performed and presented in a concise manner. Specific comments:
Abstract. It would improve readability if the gene abbreviations were spelled out.
L25 “..and has a unique cellulase expression regulation mechanism in H. insolens” More fluent to write, e.g. “..and is part of a unique cellulase expression regulation mechanism in H. insolens”
L32 “The thermophilic fungus Humicola insolens has a remarkable ability to utilize cellulose for the production of abundant enzymes to degrade cellulose and hemicellulose” This sentence has an element of circular reasoning – why not write, e.g. “The thermophilic fungus Humicola insolens produces an array of enzymes to degrade cellulose and hemicellulose”
L35 “stronger secretion capacity” Be more specific, “stronger secretion capacity of cellulases”?
L43 To improve readability, spell out the gene abbreviations.
Fig. 1 When indicating amino acids that are conserved among 4 out of 5 sequences, sometimes all residues are shown in light gray, other times only the identical residues are highlighted - be consistent.
Fig. 3. Please add a scale bar to the photos on fungal colonies. Also the medium name should be MMN, not MNN.
Fig. 4 and Fig. 6. Is there a specific reason that the experiment with non-inducing conditions was run only for 24h while that with inducing conditions was run for 6 days and harvested daily? It seems that the conclusions would have been affected had the induction experiment also been incubated only for 24h? Do you expect that no relative changes in cellulase and xylanase expression levels take place between the wild type and the transformants after 24h under non-inducing conditions?
L155 I would presume that spores were present in the cultures subjected to morphotyping, but it would appear that no observations were made concerning the potential impact of knock-out on sporulation – is this correct?
L166-180 The speculations about the underlying transcriptomic effects of the knock-out are necessary, but it is unfortunate that no transcript level profiling was included in the study. There are CDS and mRNA sequences available for endoglucanases and xylanases of H. insolens in the public domain – could you not have made use of those to design and run RT-qPCR assays?
L219 write “MMN” instead of MNN
L221 “for 24 h on PDA and YPD plates on cellophane.” Better to write “for 24 h on cellophane-coated PDA and YPD plates.”
L232 write “were” instead of “was”
Round 2
Reviewer 1 Report
The authors have taken into consideration all the comments and edits suggested by Reviewer 1 and therefore no additional edits are required.